# Antibacterial Activity of Copper Particles Embedded in Knitted Fabrics

**DOI:** 10.3390/ma15207147

**Published:** 2022-10-13

**Authors:** Remigijus Ivanauskas, Asta Bronusiene, Algimantas Ivanauskas, Antanas Šarkinas, Ingrida Ancutiene

**Affiliations:** 1Department of Physical and Inorganic Chemistry, Kaunas University of Technology, Radvilenu Str. 19, LT-50254 Kaunas, Lithuania; 2Food Institute, Kaunas University of Technology, Radvilenu Str. 19, LT-50254 Kaunas, Lithuania

**Keywords:** antibacterial activity, copper particles, ascorbic acid, *S. aureus*, *E. coli*

## Abstract

The composition and antibacterial properties of copper particles synthesized by a very simple reduction method were studied. For the preparation of particles in knitted fabrics, copper(II) sulfate was used as a precursor and ascorbic acid as a reducing natural agent. X-ray diffraction analysis showed the crystalline nature of the obtained particles. The round or oval particles and their agglomerates in knitted fabrics consisted of copper with traces of copper(I) oxide—cuprite. The element maps and energy dispersive X-ray spectra showed a high content of copper in the samples. The samples of wool and cotton knitted fabrics with copper particles had excellent antibacterial activity against gram-positive *Staphylococcus aureus* (*S. aureus*) and gram-negative *Escherichia coli* (*E. coli*) bacterial strains. The maximum zones of inhibition were 19.3 mm for *S. aureus* and 18.3 mm for *E. coli* using wool knitted fabric and 14.7 mm and 15.3 mm using cotton knitted fabric, respectively. The obtained results showed that the modified wool and cotton fabrics are suitable for use as inserts in reusable masks due to their noticeable and long-term activity against pathogenic bacteria.

## 1. Introduction

The fast global spread of the COVID-19 pandemic has undoubtedly revealed the unpreparedness of people to deal with airborne viruses and bacteria. Therefore, opportunities to create effective materials that protect against such diseases are especially relevant. Bacteria and viruses have an inherent ability to survive on frequently touched surfaces, making it easier for them to be acquired and transferred from surfaces to people. Therefore, it is relevant to solve the problems related to surface contamination with various bacteria and viruses in medical institutions and catering services or food industry companies [1,2,3]. Various disinfectants used to kill bacteria often cause allergic reactions in the human body and become an additional pollutant when released into the environment. In recent years, great attention has been paid to the search for new, more effective, and less damaging—in terms of human health and the environment—methods to kill bacteria and viruses. Nowadays there is an increasing demand for antibacterial fabrics in the form of medical clothes, protective garments, and bed spreads to minimize the chance of nosocomial infections [4]. Bacterial strains are quickly killed on copper surfaces, and copper ions on the surface play a significant role in the killing process. In laboratory tests, it has been proven that many species of bacteria, such as *Staphylococcus aureus*, *Salmonella enterica*, *Escherichia coli*, *Listeria monocytogenes*, *Campylobacter jejuni*, *Clostridium difficile*, and *Mycobacterium tuberculosis,* are killed within a few hours on copper or copper alloy surfaces [5,6,7]. The effectiveness of pure copper and copper alloy surfaces against a wide range of microorganisms has recently been demonstrated through in vitro tests [8,9,10,11,12,13,14]. With an increase in the concentration of copper, the efficiency and speed of the killing of bacteria and viruses increases as well [15]. Copper and its alloys are used to cover surfaces commonly touched by employees and patients of medical institutions: for example, door handles, various holders, parts of chairs and tables, etc. One study [16] showed that when comparing hospital waiting rooms with ordinary chairs and with chairs containing copper particles, the microbial burden in waiting rooms with the latter was reduced by 73%. In addition, the particles of copper oxides and other various copper species have been studied for their antibacterial applications [6,10,17,18,19,20,21,22,23]. CuO nanoparticles are effective, non-toxic and eco-friendly interactive materials with high antibacterial activity [17]; these particles have been widely used in medical applications due to their antioxidant, antibacterial and antifungal activities [22]. The bactericidal effect of metal and metal compound nanoparticles is attributed to their small size and high surface to volume ratio, which allow them to interact closely with microbial membranes [24]. For these reasons, the widespread use of copper and its compounds for public health is particularly important in addressing the challenges arising from the COVID-19 pandemic. One of the possible uses would be the production of long-term individual protective masks with inserts containing copper particles. Such modified materials would not only mechanically block the entry of bacteria and viruses into the human respiratory system but would effectively kill them in the entire volume of the filter material due to contact with copper particles. In this research, we present a green route for chemical synthesis of copper particles using copper(II) sulfate and L-ascorbic acid. Through this chemical synthesis, selected natural wool and cotton knitted fabrics were modified with copper particles and used as inserts for reusable face masks. To better understand the health impact of modified knitted fabrics containing copper particles, we studied the antibacterial activity against two bacterial strains: *S. aureus* and *E. coli*. In order to characterize synthesized particles, X-ray diffraction analysis (XRD), scanning electron microscopy (SEM) and energy dispersive X-ray spectroscopy (EDX) were used.

## 2. Materials and Methods 

### 2.1. Materials

Copper sulfate pentahydrate, CuSO_4_·5H_2_O, with a purity of 99% was acquired from Sigma–Aldrich (Taufkirchen, Germany) L-ascorbic acid, C_6_H_8_O_6_, with a purity of 99% was purchased from Fluka (Oslo, Norway). For this study, 100% wool fabric rib knit (180 GSM) and 100% cotton fabric interlock knit (200 GSM) produced by Lithuanian Joint Stock Company “Utenos trikotažas” with size of 40 mm × 40 mm were used. The reusable face masks made of 92% cotton and 8% polyester (model 12802) in “Utenos trikotažas” were also used.

### 2.2. Synthesis of Copper Particles in Knitted Fabrics

In a two-stage synthesis, particles containing copper were synthesized by a simple reduction method over the entire textile volume. In the first stage, a sample of wool (W) or cotton (C) knitted fabric was saturated in 0.5 mol/L solution of copper(II) sulfate at 25 °C, and in the second stage it was treated with a 0.6 mol/L solution of reducing agent (ascorbic acid) at 40 °C for 720 min. The formed copper particles remained in the used knitted fabrics after washing with distilled water, while other reaction products were removed. Then, the samples modified with copper particles were dried and used in further studies. The synthesis of copper particles embedded in wool or cotton knitted fabrics and the preparation of inserts for reusable face masks are shown in Figure 1.

### 2.3. XRD Characterization

X-ray diffraction analysis of knitted fabrics with copper particles was performed using a D8 Advance diffractometer (Bruker AXS, Karlsruhe, Germany) operating at 40 kV and a current of 40 mA. The X-ray beam was filtered with a Ni 0.02 mm filter to suppress Cu-k alfa β-radiation. The XRD patterns were recorded in a Bragg-Brentano geometry by the use of a fast counting 1-dimensional detector Bruker LynxEye (Bruker AXS, Karlsruhe, Germany) based on silicon strip technology. Specimens were scanned from 2θ = 5 to 70° at a scanning speed of 6° 1/min using a coupled two theta/theta scan type. The diffractometer was supplied together with software package DIFFRAC.SUITE. (Diffract.EVA.v.4.5 , Bruker AXS, Karlsruhe, Germany). X-ray diffractograms were processed using the software packages Crystallographica Search Match v.2.1 and Microsoft Office Excel.

### 2.4. SEM/EDX Characterization

SEM images were performed using the Scanning Electron Microscope Quanta 200 FEG (FEI, Eindhoven, The Netherlands). Samples of knitted fabrics with copper particles were imaged under a residual pressure of 80 Pa, sufficient to avoid imaging artefacts such as sample charging, commonly resulting during the high energy electron beam analysis. Energy dispersive X-ray (EDX) spectroscopy was carried out using a Bruker XFlash 4030 detector (Bruker Corporation, Billerica, MA, USA).

### 2.5. Antibacterial Activity

Fresh 18 h cultures of gram-positive bacteria *Staphylococcus aureus* ATCC 25923 and gram-negative bacteria *Echerichia coli* ATCC25922 were used. Bacterial cultures were grown in soy peptone broth (LAB 04, LAB M) at 37 °C for 24 h. After cultivation, the culture cells were mixed with a mini-shaker, and the turbidity of the suspensions was adjusted according to 0.5 McFarland’s standard (Hood, Wilkinson, & Cavanagh, 2003). Then, the suspensions were introduced into Plate Count Agar medium cooled to 47 °C, and 10 ml of the suspensions were added with a pipette into Petri dishes with a diameter of 90 mm. When the medium hardened, 8 mm diameter circles of the test knitted fabrics were placed on the surface and pressed. The plates were incubated overnight (18–24 h) at 37 °C. Circles of untreated knitted fabrics were used as negative control for bacteria at the corresponding growing conditions. After incubation, the zones of bacterial growth inhibition were measured in millimeters (including the circles of the test samples’ diameter) and analyzed for antibacterial activity. Antibacterial activity assays were repeated thrice in plate.

## 3. Results and Discussion

### 3.1. The XRD Characterization of the Formed Particles

The synthesis method of green chemistry is the best option, because it is environmentally friendly and easier compared to the methods of traditional chemistry [25]. The green synthesis can be carried out in mild reaction conditions using a reducing natural agent, for example, ascorbic acid and copper(II) ion solution [25,26].

In ref. [27], an aqueous solution of ammonia was used to control the pH value; therefore, copper sulfate first reacted with ammonia to form insoluble copper(II) hydroxide, and then Cu(OH)_2_ was reduced using ascorbic acid, and the particles of copper were obtained. In our work, as in ref. [25], copper(II) sulfate first dissociated to Cu^2+^ and SO_4_^2–^; then, ascorbic acid was oxidized, and copper(II) ion was reduced to Cu. The color of the reaction medium changed from light blue (Cu^2+^) to light green (Cu^+^) and finally to brown (Cu^0^). The XRD results in refs. [25,26,27] show that the formed particles consisted of copper (JCPDS number 4–836). It was also shown in ref. [26] that at the early stages of the reaction between copper ions and ascorbic acid, copper(I) oxide, Cu_2_O (JCPDS number 5–667), predominates, which gradually turns into copper after 180 min. The authors explain this by the fact that Cu particles are formed as a result of a two-stage reduction, with the formation of Cu_2_O as an intermediate product. In order to determine the composition of our synthesized particles, phase analysis of the particles was performed by X-ray diffraction analysis. The data obtained were compared with JCPDS data and assigned to *cubic* copper Cu (4–836) and *cubic* cuprite Cu_2_O (5–667). The data in the table show that the experimentally determined interplanar spacings are close to the JCPDS data (Table 1).

As can be seen from the data presented in the top of Figure 2, only the Cu (4–836) peaks at 2ϴ = 43.30° and 50.44° clearly dominate in the X-ray diffractogram. At that time, only two negligible peaks of Cu_2_O (5–667) were observed at 2ϴ = 29.54° and 36.39°. Therefore, it can be stated that the synthesized particles consist of copper with traces of copper(I) oxide-cuprite.

The data of X-ray diffraction analysis was also confirmed by the color of the formed particles-brown particles, characteristic of elemental copper.

### 3.2. SEM/EDX Characterization of Knitted Fabrics with Formed Copper Particles

The morphology of modified samples and the structure and distribution of particles in knitted wool or cotton fabrics was evaluated using scanning electron microscopy. The round or oval spherical monodispersed copper particles and their agglomerates could be observed in SEM images (Figure 3) at different magnification.

It is known [25] that green synthesized Cu particles have a spherical morphology.

The chemical elemental composition of the modified samples was analyzed using a scanning electron microscope with an EDX detector (Bruker Corporation, Billerica, MA, USA); chemical element maps were also made. The presented element maps and EDX spectra (Figure 4) showed a high content of copper in the studied samples.

The results of this analysis confirmed the results of X-ray diffraction analysis, since the copper peaks predominated in the diffraction pattern (Figure 2), indicating the formation of copper particles. A small amount of sulfur was also observed in EDX spectra. This can be explained by the fact that a small amount of adsorbed sulfate ions remained in the samples.

### 3.3. Antibacterial Activity of Copper Particles in Knitted Fabrics

The antibacterial activities of the untreated and modified samples of wool and cotton knitted fabrics were studied by zone of inhibition tests. Two the most common pathogenic bacteria, *S. aureus* (gram-positive) and *E. coli* (gram-negative), were selected for the study of the killing activity of modified wool and cotton. The antibacterial properties of copper, copper alloys and various copper compounds were studied many works [1,3,11,15,16,17,18,19,20,21,22,23,25]. *Staphylococci* were identified as an important cause of both nosocomial and community-acquired infections [16,21,28]. *S. aureus* bacteria is naturally found in the nasopharynx and in the nose and skin and can be spread by contact; healthy carriers or sick people can pass staphylococci to others by airborne droplets, as well as through contaminated hands or a wide variety of household items. *E. coli* is a popular bacteria that is considered to be a common inhabitant of the human intestinal tract and has the ability to grow both with and without oxygen [21]. Importantly, copper-based particles can kill gram-negative bacteria (*E. coli*) and gram-positive bacteria (*S. aureus*) and can be used to treat surgical wounds, burns, and diabetic foot infections [29].

The antibacterial activities of modified wool and cotton knitted fabrics were examined though a comparison with the untreated fabrics. In control experiments on the antimicrobial behavior of wool and cotton knitted fabrics against the tested bacteria, it was found that untreated samples did not show antibacterial activity because the diameter of the inhibition zone was 0 mm (Figure 5, bottom). Samples of knitted fabrics with copper particles had an excellent antibacterial activity because of a very large diameter of the inhibition zone (Figure 5, top). The samples of modified wool had the largest zone of *S. aureus* bacteria inhibition (19.3 ± 0.6 mm), and they also had a large inhibition zone (18.3 ± 0.6 mm) against *E. coli* bacteria. The determined diameters of the inhibition zone indicated that the modified wool samples inhibited *S. aureus* more effectively than *E. coli*, as in ref. [3], where samples of copper-based membranes had an inhibition zone diameter of over 20 mm for *S. aureus* bacteria and 15 mm for *E. coli* bacteria. The samples of modified cotton knitted fabrics also had very good antibacterial activity, but the diameters of inhibition zones against both bacteria were smaller. The diameter of the inhibition zone was 14.7 ± 0.6 mm for *S. aureus* and 15.3 ± 0.6 mm for *E. coli* bacteria.

These differences in the antibacterial activity of the modified wool and cotton samples can be explained by differences in the weight changes of the wool and cotton knitted fabrics before and after treatment. The weight changes of the modified samples were about 15% for wool and about 9% for cotton samples. Hence, a larger amount of physically adsorbed copper particles was in the samples of modified wool. A higher adsorption capacity of Cu^2+^ ions in wool fibers compared to other cations was noted by Monier et al. [30]. Due to wool swelling, a large flow of Cu^2+^ ions could be adsorbed not only on the surface, but throughout the entire volume of wool knitted fabric. Consequently, more copper particles were formed in the samples of modified wool.

The effect of the concentration of solutions on antibacterial properties was also studied. Samples of wool and cotton knitted fabrics were exposed to solutions of different concentrations of copper(II) sulfate (0.25 M; 0.5 M and 0.75 M) and ascorbic acid (0.3 M; 0.6 M and 0.9 M). From the data presented in Table 2, it can be seen that the samples had worse antibacterial activity when using solutions with lower concentrations, as the inhibition zones for *S. aureus* bacteria were much smaller, and an increase in the concentration of solutions did not lead to a significant improvement in the bactericidal properties of the samples; thus, the optimal variant was chosen, and for further studies 0.5 M CuSO_4_ and 0.6 M C_6_H_8_O_6_ solutions were used.

In this study, the long-term bactericidal effect of copper particles was also investigated. Since bacteria are often transmitted by contact, and *S. aureus* bacteria are also transmitted by airborne droplets, inserts of modified knitted fabrics made from natural wool or cotton fibers were placed in the reusable masks. The masks of the first group had an insert (MI1, MI2 and MI3) made of wool knitted fabric with copper particles (W). The masks of the second group had an insert (MI4, MI5 and MI6) made of cotton knitted fabric with copper particles (C). A total of 18 masks were prepared, of which 9 were with modified wool inserts and the other 9 were with modified cotton inserts. These masks were worn by the authors of this article for 2, 4 and 8 h for 5 working days, namely, each mask was worn for 5 days for a different set number of hours. Participants wore masks with inserts at work, on the street, in shops and on public transport, striving to be as diverse as possible. After wearing, the bactericidal activity of the inserts was examined and compared with the antibacterial properties of the non-worn corresponding samples. The results are presented in Table 3 and Figure 6 and Figure 7.

The results obtained (Table 3, Figure 6) showed that after wearing the masks for 2 hours for 5 days, the bactericidal activity of the modified wool knitted fabric was slightly reduced against *S. aureus* and almost unchanged against *E. coli* bacteria. An insignificant decrease in the diameter of the zones of inhibition of bacteria was observed with increased wear time, but the samples showed excellent antibacterial activity even after 8 h of wear. The results in Table 3 and Figure 7 show that after 2 h of wearing for 5 days, the bactericidal effect of the modified cotton knitted fabric against *S. aureus* and *E. coli* decreased slightly; in all cases, a decrease in the diameter of the zones of inhibition of bacteria was observed. There was a negligible decrease in the diameter of the bacterial inhibition zones with increased wear time, and the samples still demonstrated good antibacterial activity.

In summary, this research was designed to evaluate the possibility of the use of natural knitted fabrics with copper particles as inserts in reusable masks. The obtained results showed that the modified wool and cotton fabrics are suitable for use in long-term protection measures due to their noticeable and long-term activity against pathogenic bacteria.

## 4. Conclusions

L-ascorbic acid (C_6_H_8_O_6_) is green and a suitable reducing agent for the formation of copper particles by the use of copper(II) sulfate as a precursor. X-ray diffraction analysis of the formed round or oval spherical particles revealed that copper (JCPDS number 4–836) predominates, and this result was confirmed by energy dispersion X-ray spectroscopy. The antibacterial efficacy of untreated wool and cotton knitted fabrics and those modified with copper particles fabrics was tested against gram-positive bacteria *Staphylococcus aureus* and gram-negative bacteria *Escherichia coli*. The modified wool fabric was found to have the best antibacterial properties against *S. aureus* bacteria; the diameter of the inhibition zone reached 19.3 mm, and the maximum zone of inhibition for *E. coli* was 18.3 mm. The maximum zones of inhibition were 14.7 mm for *S. aureus* and 15.3 mm for *E. coli* using cotton fabric. The presence of copper particles embedded in knitted fabrics ensured excellent and long-term antibacterial activity against gram-positive and gram-negative bacteria; hence, it is possible to use modified wool and cotton fabrics in the production of inserts for reusable masks and infection prevention.

## Figures and Tables

**Figure 1 materials-15-07147-f001:**
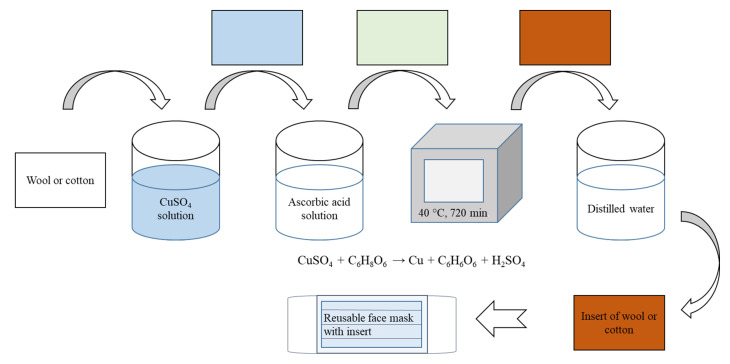
Schematic diagram of the synthesis of copper particles and the preparation of inserts for reusable face masks.

**Figure 2 materials-15-07147-f002:**
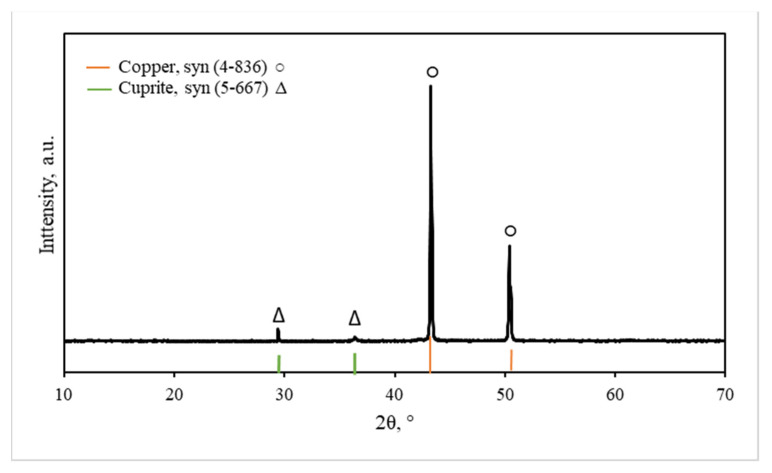
X-ray diffractogram of the particles formed: X-ray experimental diffractogram (top), X-ray diffractograms of Cu (4–836) and Cu_2_O (5–667) according to JCPDS data (bottom).

**Figure 3 materials-15-07147-f003:**
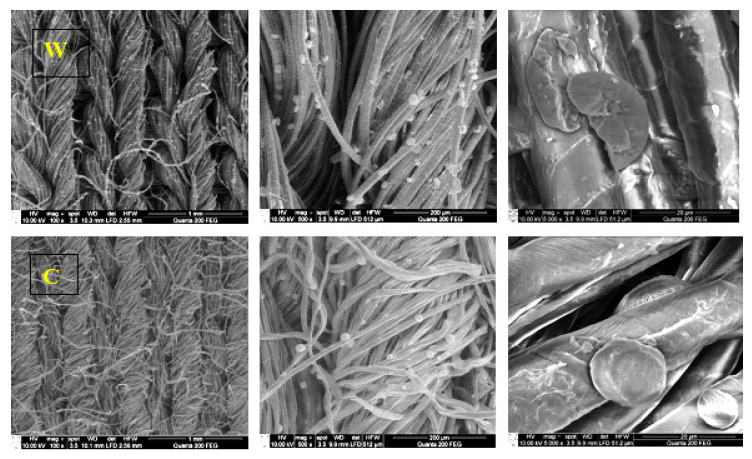
SEM images of W (**top**) and C (**bottom**) at 100×, 500× and 5000× magnification and particle distribution.

**Figure 4 materials-15-07147-f004:**
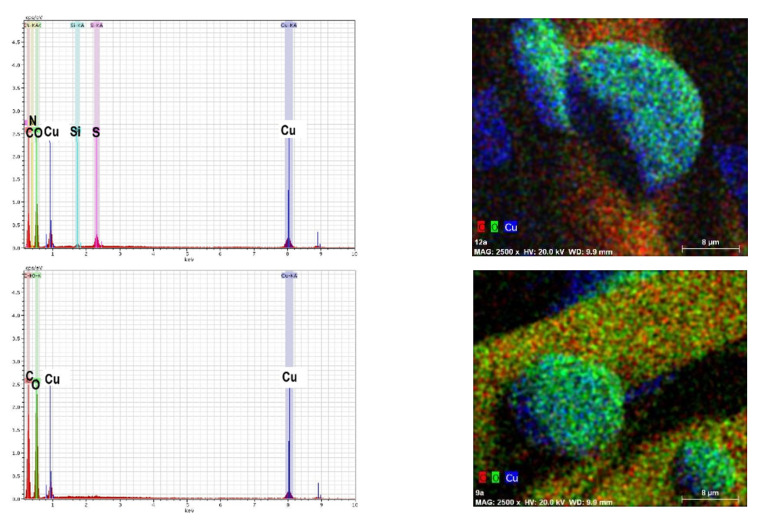
EDX spectra of the modified samples W (**top**) and C (**bottom**) and their chemical element maps (magnification 2500×).

**Figure 5 materials-15-07147-f005:**
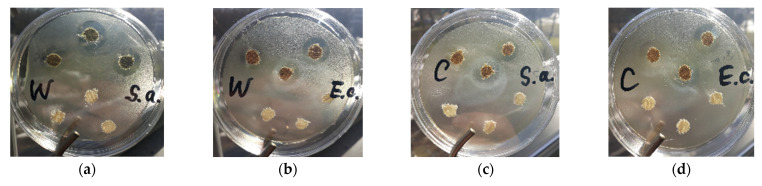
Photographs of untreated (bottom) and modified (top) samples of wool (W) and cotton (C) knitted fabrics against *S. aureus* (**a**,**c** photos) and *E. coli* (**b**,**d** photos) bacteria.

**Figure 6 materials-15-07147-f006:**
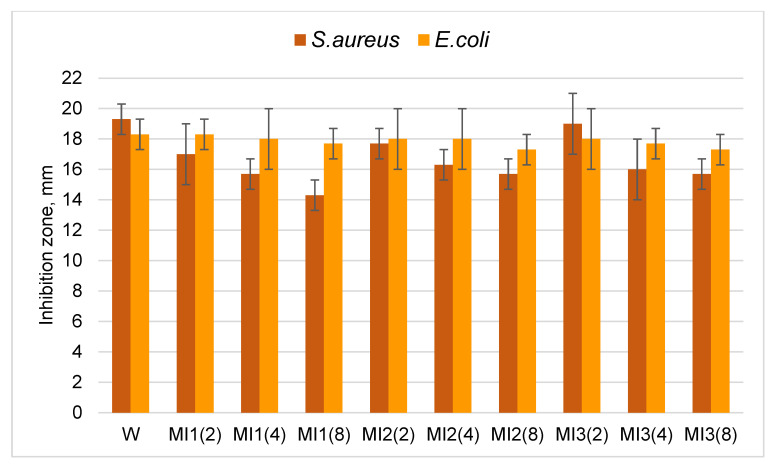
Diameters (mm) of inhibition zones against *S. aureus* and *E. coli* bacteria of modified wool knitted fabric inserts used in reusable masks. W is the non-worn sample; MI1 (2), MI2 (2) and MI3 (2) after 2 h of wearing; MI1 (4), MI2 (4) and MI3 (4) after 4 h of wearing; MI1 (8), MI2 (8), and MI3 (8) after 8 h of wearing.

**Figure 7 materials-15-07147-f007:**
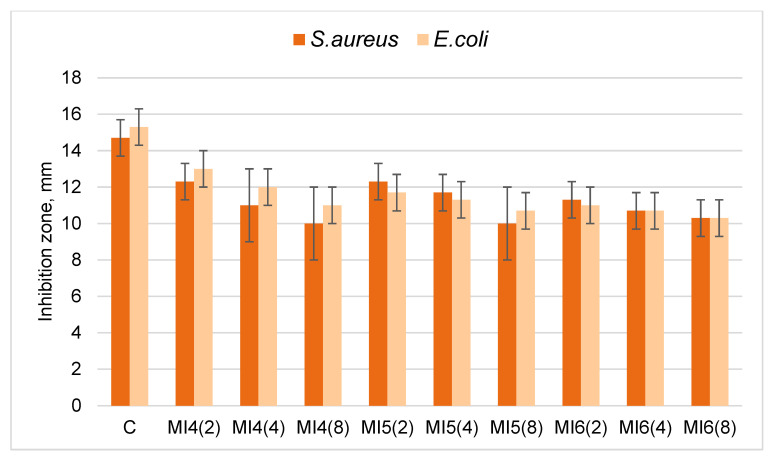
Diameters (mm) of inhibition zones against *S. aureus* and *E. coli* bacteria of modified cotton knitted fabric inserts used in reusable masks. C is the non-worn sample; MI4 (2), MI5 (2) and MI6 (2) after 2 h of wearing; MI4 (4), MI5 (4) and MI6 (4) after 4 h of wearing; MI4 (8), MI5 (8), and MI6 (8) after 8 h of wearing.

**Table 1 materials-15-07147-t001:** Comparison of experimentally determined interplanar spacings with JCPDS data.

2θ, (Degree)	Interplanar Spacing (d), Å	
	Experimental Data	JCPDS Data
29.54	3.029	3.033
36.39	2.466	2.465
43.30	2.088	2.088
50.44	1.808	1.808

**Table 2 materials-15-07147-t002:** Bactericidal activity of modified wool and cotton samples against *S. aureus.*

Concentrations of Solutions	Inhibition Zone, mm	
	Samples of Modified Wool	Samples of Modified Cotton
0.25 M CuSO_4_ and 0.3 M C_6_H_8_O_6_	15.7 ± 0.6	12.3 ± 0.6
0.5 M CuSO_4_ and 0.6 M C_6_H_8_O_6_	19.3 ± 0.6	14.7 ± 0.6
0.75 M CuSO_4_ and 0.9 M C_6_H_8_O_6_	21.0 ± 1.0	15.3 ± 0.6

**Table 3 materials-15-07147-t003:** Antibacterial results for non-worn samples (W and C) and the mask inserts.

Sample	Inhibition Zone, mm		Sample	Inhibition Zone, mm	
	*S. aureus*	*E. coli*		*S. aureus*	*E. coli*
W	19.3 ± 0.6	18.3 ± 0.6	C	14.7 ± 0.6	15.3 ± 0.6
MI1(2)	17.0 ± 1.0	18.3 ± 0.6	MI4(2)	12.3 ± 0.6	13.0 ± 1.0
MI1(4)	15.7 ± 0.6	18.0 ± 1.0	MI4(4)	11.0 ± 1.0	12.0 ± 1.0
MI1(8)	14.3 ± 0.6	17.7 ± 0.6	MI4(8)	10.0 ± 1.0	11.0 ± 1.0
MI2(2)	17.7 ± 0.6	18.0 ± 1.0	MI5(2)	12.3 ± 0.6	11.7 ± 0.6
MI2(4)	16.3 ± 0.6	18.0 ± 1.0	MI5(4)	11.7 ± 0.6	11.3 ± 0.6
MI2(8)	15.7 ± 0.6	17.3 ± 0.6	MI5(8)	10.0 ± 1.0	10.7 ± 0.6
MI3(2)	19.0 ± 1.0	18.0 ± 1.0	MI6(2)	11.3 ± 0.6	11.0 ± 1.0
MI3(4)	16.0 ± 1.0	17.7 ± 0.6	MI6(4)	10.7 ± 0.6	10.7 ± 0.6
MI3(8)	15.7 ± 0.6	17.3 ± 0.6	MI6(8)	10.3 ± 0.6	10.3 ± 0.6

## Data Availability

Not applicable.

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
