# Peer review of "Antibacterial Activity of Copper Particles Embedded in Knitted Fabrics"

_materials, 2022, doi:10.3390/ma15207147_

Round 1
Reviewer 1 Report
This manuscript reported a green route for chemical synthesis of copper particles using copper sulfate and L-ascorbic acid. Through this chemical synthesis, selected natural wool and cotton knitted fabrics were modified with copper particles and used as inserts for reusable face masks. The manuscript has been well written. However, after reading through the manuscript, I have to suggest MAJOR revision with detailed comments as follows:
1. Introduction section: Some introductions about COVID-19 pandemic should be revised. Because the theme in manuscript is not related to the evaluation of anti-coronavirus for Copper particles embedded in knitted fabrics. This works mainly investigated the antibacterial activity of Cu.
2. Statistical analysis on experimental data was lacking. How is the statistically significant difference of antibacterial results. Data should express as mean ±â€¯standard deviation.
3. For “synthesis of copper particles in knitted fabrics”, how does reaction time affect the formation of Cu after adding reducing agent? In this work, 720 min was used, why? Different reaction time should be investigated.
4. The phrases like “In [27]” should be corrected.
5. Why does Cu modified wool samples have better antibacterial ability of S. aureus than Cu modified cotton knitted fabrics? This point should be discussed.
6. The bar and significant differences should be added in Fig. 6 and 7.
Author Response
Thank you for reviewing our manuscript and improving its quality. We revised it according to the comments

Reviewer 2 Report
Minor comments to be addressed
1. How much copper were embedded on the cotton/wool
2. author has to give quantitative value of the copper embedded on the material.
3. use some commercial drug for antibacterial activity comparisions.
4. how the author decided the quantity of CUSO4 to be used in this paper.
5. statistic data for the results in the figures are missing
Author Response

(The authors gave the same response as above.)

Reviewer 3 Report
In this paper, antibacterial activity of copper particles embedded in knitted fabrics were studied. The authors have studied the bacterial growth of inhibition in a systematic manner. The paper is interesting, well presented and written. However, the manuscript can be published in the journal after the following revision.
Comments:
1. Figure 1 is looking very poor. It should be redrawn by explaining the complete mechanism.
2. How copper particles are bonded on wool or cotton during the synthesis. Is it physical adsorption or else? Please explain the mechanism behind it.
Author Response

(The authors gave the same response as above.)

Round 2
Reviewer 1 Report
The revised manuscript can be published.
Reviewer 3 Report
The authors have improved the manuscript in the revised version. The paper can be published in the journal.